



# Are agricultural plastic covers a source of plastic debris in soil? A first screening study

Zacharias Steinmetz[1], Paul Löffler[1], Silvia Eichhöfer[1], Jan David[1], Katherine Muñoz[2], and Gabriele E. Schaumann[1]

[1]iES Landau, Institute for Environmental Sciences, Group of Environmental and Soil Chemistry, University of Koblenz–Landau, Fortstraße 7, 76829 Landau, Germany
[2]iES Landau, Institute for Environmental Sciences, Group of Organic and Ecological Chemistry, University of Koblenz–Landau, Fortstraße 7, 76829 Landau, Germany

**Correspondence:** Gabriele E. Schaumann (schaumann@uni-landau.de)

**Abstract.** Agricultural plastic covers made from polyethylene (PE) and polypropylene (PP) offer increased yields and an improved crop quality. However, such covers are suspected of partially breaking down into smaller debris and thereby contributing to soil pollution with microplastics. To scrutinize this, we randomly sampled 240 topsoil cores (0–5 cm) from eight fields covered with fleeces, perforated foils, and plastic mulches for less than two years. Samples from the field periphery (50 m perimeter) served as reference. Visual plastic debris >2 mm was analyzed by Fourier transformed infrared spectroscopy with attenuated total reflection (FTIR–ATR). Smaller, soil-associated plastic debris was dispersed from 50 g of fine soil (≤2 mm) using sodium hexametaphosphate solution and density-separated with saturated $NaCl$ solution. The collected PE, PP, and polystyrene (PS) debris was selectively dissolved in a mixture of 1,2,4-trichlorobenzene and *p*-xylene at 150 °C and quantified by pyrolysis-gas chromatography/mass spectrometry (Py-GC/MS). We counted six PE and PS fragments >2 mm in two out of eight fields. By contrast, Py-GC/MS analysis revealed PE, PP, and PS contents >1 $\mu g\,g^{-1}$ in seven fields (17 % of all samples). In three fields, PE levels of 3–35 $\mu g\,g^{-1}$ were associated with the use of thinner and less durable perforated foils (40 μm thickness). This was slightly more pronounced at field edges where the plastic covers are turned and weighted down. By contrast, 50 μm thick PE films were not indicated to emit any plastic debris. PP contents of 5–10 $\mu g\,g^{-1}$ were restricted to single observations in the field centers of three sites. On one site, we found expanded PS particles >2 mm that concurred with elevated PS levels (8–19 $\mu g\,g^{-1}$) in the fine soil. Both PP and PS were distributed indistinctly across sites so that their source remained unresolved. In addition, the extent to which plastic contents of up to 7 $\mu g\,g^{-1}$ in the field periphery of some sites were attributed to wind drift from the covered fields or from external sources needs to be investigated in future studies. Yet, our results suggest that the short-term use of thicker and more durable plastic covers should be preferred to limit plastic emissions and accumulation in soil.

## 1 Introduction

The use of plastic covers has become common agricultural practice for improving yields and crop quality, managing harvest times, and increasing pesticide and water use efficiency (Lamont, 1993; Steinmetz et al., 2016). The most used materials are



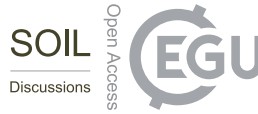

polyethylene (PE) films and polypropylene (PP) fleeces of various thicknesses made to last for up to 10 years (Bertling et al., 2021). However, wind, heavy machinery, or UV irradiation are likely to disintegrate parts of the covers into debris smaller than

1–5 mm (Scarascia-Mugnozza et al., 2011), termed microplastics (Hartmann et al., 2019). In recent years, this supposition has raised a discussion about agricultural plastic covers acting as a potential source of plastic debris in the terrestrial environment and particularly in soil (Steinmetz et al., 2016; Hurley and Nizzetto, 2018). Yet, the actual contribution of agricultural plastic covers to soil pollution with plastic debris has remained incompletely understood and rarely discriminated from other potential sources like aerial deposition or littering.

These knowledge gaps are probably because the few studies that have analyzed plastics in and on soil so far mostly relied on optical detection by Fourier transformed infrared (FTIR) spectroscopy or visual microscopy. Both techniques deliver particle counts, are relatively sensitive to matrix interferences, and thus require extensive sample preparation when applied to hetero­geneous matrices with a similar particle structure to the plastic particles of interest (Thomas et al., 2020). For those reasons, Piehl et al. (2018) and Harms et al. (2021) excluded plastic debris <1 mm from their FTIR analysis of agricultural topsoil

(0–5 cm). The investigated sites were not covered with plastic, yet the soil contained 0.3 to 6 particles $kg^{-1}$ of 1–5 mm size. These findings contrast Zhang and Liu (2018) who detected 95 % of plastic debris <10 mm (up to 40000 particles $kg^{-1}$) in the size fraction of 0.05–1 mm after more than 25 years of permanent greenhouse cultivation. Topsoil previously covered with plastic revealed plastic counts of 60 up to 1000 particles $kg^{-1}$ correlating with the 5–24 years of continuous plastic coverage (Huang et al., 2020). PE and PP are typically found the most (Harms et al., 2021; Kim et al., 2021). However, there are still

studies which neither state particle sizes and analysis cutoffs nor assess the polymer composition of the retrieved particles (for example Zhang et al., 2018; Beriot et al., 2021). Moreover, mass-based information is still missing but urgently needed for the monitoring and regulation of plastics in the environment.

With this study, we aimed to better understand the mass distribution of plastic debris associated with fine soil (≤2 mm) and to scrutinize the extent to which agricultural plastic covers emit plastic debris into their surrounding. In contrast to other

studies, we screened fields covered with plastic for less than two years, which reflects typical land use and crop rotations in Germany (Harms et al., 2021). To this end, we randomly sampled topsoil within and around eight commercially managed agricultural fields covered with fleeces, perforated foils, and plastic mulches. PE, PP, and polystyrene (PS) debris ≤2 mm was quantified by solvent-based pyrolysis-gas chromatography/mass spectrometry (Py-GC/MS) (Steinmetz et al., 2020). To better account for the heterogeneous distribution of plastic debris in soil, we further refined and validated a new sample preparation

procedure involving soil aggregate dispersion and density separation. Our analyses were complemented by FTIR spectroscopy with attenuated total reflection (ATR) for plastic debris >2 mm. We hypothesized that a directed gradient of plastic debris from the field center to its periphery (50 m field perimeter) supports the assumption of plastic covers contributing to an increased soil pollution with plastic debris. On the contrary, an undirected gradient would suggest another source of pollution such as littering. A uniform distribution may be an indicator for aerial deposition. In addition, we expected field margins (5 m

perimeter) to be hotspots for plastic debris due to mechanical stress subjected to the plastic covers by weighting them down with soil or sandbags.





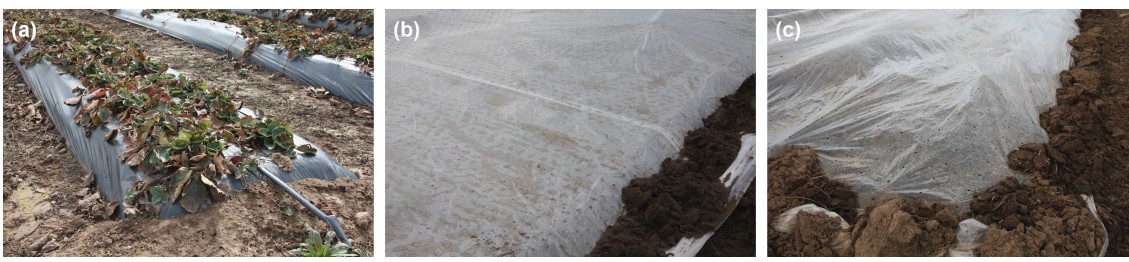

**Figure 1.** Exemplary photographs of site 3, 4, and 8 field edges covered with (a) mulch, (b) fleece and perforated foil, and (c) perforated foil, respectively.

## 2  Methods

### 2.1  Study area

The screening study was conducted in cooperation with local farmers on commercially managed horticultural fields in the Palatinate region in southwestern Germany. The study area has a mild and dry climate with mean annual temperatures of 10–13 °C and a total annual precipitation of 600±100 mm (Agrarmeteorologie Rheinland-Pfalz, 2020).

Sites 1–3 were located near Offenbach an der Queich (49° 12' N, 8° 11' E) and cultivated with strawberries (*Fragaria × ananassa*). Site 1 was fully covered with white fleece (100 μm thickness) and overlain by an additional 40 μm thick perforated foil (750 punch holes $m^{-2}$) for one growing season (last four months). Site 2 had plastic-mulched ridges (black, 50 μm thickness) and bare furrows established two years ago. On top of it, the complete field was covered with white fleece (100 μm thickness) for the growing season. Site 3 was mulched like site 2 but without any additional fleece cover (Fig. 1a).

Sites 4 and 5 were located near Schifferstadt (49° 24' N, 8° 21' E) and cultivated with lettuce (*Cichorium endivia*) and cabbage (*Brassica oleracea* var. *gongylodes*), respectively. Both fields were completely covered with white fleece (40 μm thickness) and a top layer of white perforated foil (50 μm thickness, 750 punch holes $m^{-2}$, Fig. 1b) for the growing season.

Sites 6–8 were situated in Landau in der Pfalz (49° 11 N, 8° 10' E). The rhubarb cultivation (*Rheum rhabarbarum*) was fully covered with white perforated foil for the growing season. The foil on site 6 had 250 punch holes $m^{-2}$ and was 50 μm thick. On site 7, the number of punch holes was similar to site 6, but the film thickness was only 40 μm. By contrast, the foil on site 8 (40 μm thickness) had 750 punch holes $m^{-2}$ (Fig. 1c).

### 2.2  Sampling strategy

To systematically screen these agricultural fields for plastic debris, each site was subdivided into four transects as shown in Fig. 2. The field center and the inner field edge, defined as a 5 m strip around the field center, were cultivated and covered with plastic film. In these transects, plant rows (ridges) and track rows (furrows) were sampled separately. The outer field margin was marked by the 5 m perimeter around the cultivation. The field periphery (50 m perimeter around the cultivation) served as reference.





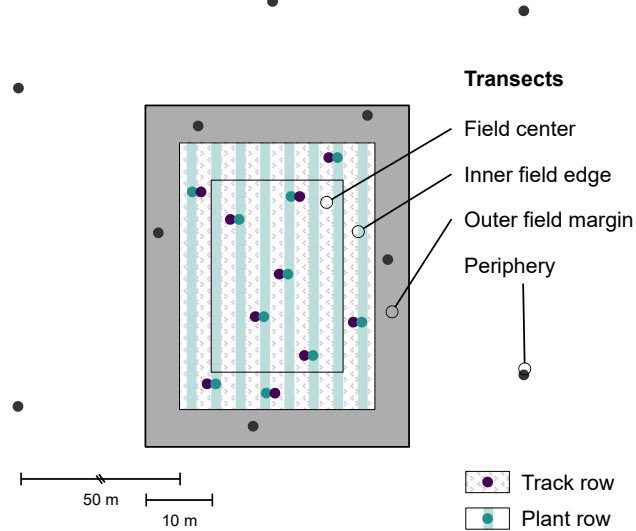

**Figure 2.** Sampling scheme for one exemplary site; soil samples (filled dots, $n = 5$ per transect) were randomly selected. At the cultivated field centers and inner field edges, plant and track rows were sampled separately. The outer field margin and the field periphery were uncultivated.

Prior to retrieval of the plastic covers in spring 2018, small portions of the plastic cover were grab-sampled for subsequent characterization. Soil samples were taken <1 week after retrieval of the plastic covers. The sampling spots were predefined by projecting a $1 \times 1$ m grid onto each transect and randomly selecting five squares without replacement (30 per site). At each of these spots, topsoil (0–5 cm depth) was sampled using a stainless steel core cutter (5 cm diameter). The soil cores were immediately transferred to uncoated paper bags and air-dried therein to reduce the risk of contamination.

**2.3    Soil characterization and chemical analysis of plastic covers**

The soil texture on each site was estimated from composite subsamples using the hydrometer method described in ASTM D422-63 (2007). The electric conductivity (EC) and pH were measured in deionized water and $0.01$ M $CaCl_2$ aqueous solution, respectively. Soil carbon and nitrogen were determined by dry combustion elemental analysis (Vario MICRO Cube, Elementar, Germany).

The grab-sampled plastic covers were characterized by qualitative thermodesorption (TD)- and Py-GC/MS, differential scanning calorimetry (DSC), thermogravimetry/mass spectrometry (TGA/MS), and FTIR–ATR analysis. TD- and Py-GC/MS were applied to assess volatile additives or other polymer-associated compounds as well as the overall polymer composition of the agricultural plastic covers. To this end, a $1 \times 1$ mm piece was cut out of each plastic cover and placed into a pyrolyzer quartz tube of a Pyroprobe 6150 filament pyrolyzer (CDS Analytical, Oxford, United States) coupled with a Trace GC Ultra

with DSQII mass spectrometer (MS) (Thermo Fisher Scientific, Bremen, Germany). For the TD, the pyrolyzer interface was flash heated ($10$ K ms$^{-1}$) to $300$ °C for $15$ s to volatilize any polymer-associated compounds. A passivated transfer line





(350 °C) transferred the volatiles to the split/splitless injector (300 °C, split ratio 1:75) of the GC/MS system. The compounds were chromatographically separated in a 1.3 mL min$^{-1}$ He flow on a 30 m × 0.25 mm capillary column (5 % phenyl-arylene, 95 % dimethylpolysiloxane, 0.25 μm film thickness, ZB-5MS, Phenomenex, Aschaffenburg, Germany). The oven program was: 40 °C (2 min hold), 8 K min$^{-1}$ ramp to 300 °C (5 min hold). The GC/MS transfer line was kept at 280 °C, and the MS ion source (70 eV) was heated to 230 °C. The MS monitored *m/z* 50–280 at a scan rate of 500 s$^{-1}$. After TD, the sample was pyrolyzed at 750 °C for 15 s applying the same GC/MS settings. All chromatograms were evaluated using OpenChrom, version 1.4.0.202103172155 (Wenig and Odermatt, 2010), with the NIST08 database for peak identification.

DSC and TGA/MS measurements were conducted in accordance with David et al. (2018). In brief, DSC was applied between −50 and 250 °C (10 K min$^{-1}$ ramp, 50 mL min$^{-1}$ N$_2$ flow, Q1000, TA Instruments, New Castle, US) to determine the melting and crystallization temperatures of the agricultural plastic films. For the determination of polymer degradation onsets and evolved gases, plastic samples were subjected to TGA/MS analysis (STA 449 F3 Jupiter with QMS 403 C Aëolos, Netzsch, Selb, Germany). The heating ramp was 5 K min$^{-1}$ from 40 to 1000 °C under a 20 mL min$^{-1}$ Ar flow. The MS monitored *m/z*s 12–32 and 44 at a dwell time of 1 s and *m/z*s 33–154 at a dwell time of 5 s.

Complementary FTIR–ATR analyses were performed at 4000–650 cm$^{-1}$ (4 cm$^{-1}$ resolution) using a Cary 630 spectrometer (Agilent, Santa Clara, California, US). Peaks were identified with Open Specy, version 0.9.2 (Cowger et al., 2021).

### 2.4  Soil sample preparation and visual pre-screening

All soil cores were homogenized and sieved to fine soil (≤2 mm) as suggested by Thomas et al. (2020). Visual plastic items retained by the sieve (>2 mm) were manually picked, photographed (Leica S9i, Wetzlar, Germany), and analyzed via FTIR–ATR as described in the previous section.

Plastic debris ≤2 mm were density-separated from the soil matrix using saturated NaCl solution. To this end, 50 g of fine soil were first weighted into 1 L separation funnels with polytetrafluoroethylene (PTFE) stop cock (Carl Roth, Karlsruhe, Germany) and agitated at 150 rpm with 125 mL of sodium hexametaphosphate (40 g L$^{-1}$, CAS 68915-31-1, ≥99 % purity, Carl Roth, Karlsruhe, Germany) for 2 h to disperse any soil aggregates. In a second step, 90 g of NaCl (CAS 7647-14-5, ≥99.8 % purity, Carl Roth, Karlsruhe, Germany) and 125 mL of ultra-pure water were added to obtain a density solution of 1.2 g cm$^{-3}$. The mixture was shaken for another 2 h and left for sedimentation for at least 16 h. The sedimented soil was released from the separation funnel by gentle stirring of the suspension using the curved end of a bicycle spoke. Afterwards, the supernatant was collected in pleated cellulose filters (Whatman 589/2, 4–12 μm particle retention, GE Healthcare, Buckinghamshire, UK). The filters cakes were transferred into glass culture tubes (16 × 100 mm, GL18, VWR, Darmstadt Germany) and dried at 60 °C.

Based on Steinmetz et al. (2020), the culture tubes were topped off with 8 mL of a 1:1-mixture (v+v) of *p*-xylene (CAS 106-42-3, >98.0 % purity, Fluka Analytical, München, Germany) and 1,2,4-trichlorobenzene (TCB, CAS 120-82-1, 99 % purity, Alfa Aesar, Kandel, Germany). In addition, the mixture contained 100 mg L$^{-1}$ butylated hydroxytoluene (BHT, CAS 128-37-0, ≥99 %, Merck, Darmstadt, Germany) to prevent polymer oxidation. The tubes were sealed with a PTFE packing (Carl Roth, Karlsruhe, Germany), vortexed, and heated at 150 °C for 1 h to facilitate extraction of the polymer analytes from the filter cake. After cooled down to room temperature, the supernatant was spiked with deuterated PS (PS-d5, PolymerSource,





Quebec, Canada) for internal standardization using positive displacement pipettes with glass capillaries (Transferpettor micro, Brand, Wertheim, Germany). The extracts were stored in 2 mL ND9 glass vials with inserts and PTFE-sealed caps (Wicom, Heppenheim, Germany).

## 2.5 Quantification of plastic debris in soil

PE, PP, and PS debris in fine soil ($\leq 2$ mm) were quantified via Py-GC/MS as detailed in Steinmetz et al. (2020). In brief, 2 μL sample aliquots were injected into pyrolyzer quartz tubes equipped with two Whatman QM-A microfiber filter disks (Kent, United Kingdom) using a 10 μL syringe with PTFE plunger (Hamilton 1701 N with 26s gauge, Bonaduz, Switzerland). The Py-GC/MS analysis was performed as described in Section 2.3. However, the pyrolyzer interface was first held at 300 °C to purge remaining solvents and volatiles on-line. After 3 min, the sample was flash pyrolyzed ($10 \, \text{K} \, \text{ms}^{-1}$) at 700 °C for 15 s

and transferred to the GC/MS system. The MS selectively monitored $m/z$s 70 and 126 for the PP pyrolysate 2,4-dimethyl-1-heptene (2,4Me9:1(1), RI 841), $m/z$s 104 and 118 for the PS pyrolysates styrene (Sty, RI 895) and $\alpha$-methylstyrene ($\alpha$MeSty, RI 981), respectively, and $m/z$s 82 and 95 for PE $n$-alkadienes like 1,21-docosadiene (22:2(1,21), RI 2187). The internal standard styrene-d5 (Sty-d5, RI 892) was acquired at $m/z$ 109.

## 2.6 Method validation and quality control

The reference polymers used for external standardization and recovery experiments were analytical grade PE beads (CAS 9002-88-4, 500 μm average particle size) from Alfa Aesar, Kandel, Germany, PP fragments (CAS 9003-07-0, isotactic, $\leq$ 1000 μm) from Aldrich Chemistry, Taufkirchen, Germany, and PS beads (CAS 9003-53-6, 250 μm average particle size) from Goodfellow, Huntingdon, United Kingdom (see Steinmetz et al., 2020, for details).

The Py-GC/MS system was calibrated weekly against external standards (5–200 $\text{μg} \, \text{mL}^{-1}$ PE, PP, and PS dissolved in xy-

lene/TCB at 150 °C) following the protocol by Steinmetz et al. (2020). Calibration curves were evaluated for signal sensitivity (slope) and linearity (adj. $R^2$). Daily sample measurements were bracketed with 100 $\text{μg} \, \text{mL}^{-1}$ standards to correct for inter-day variations. The intra-day repeatability was determined by consecutive injections of 100 $\text{μg} \, \text{mL}^{-1}$ standards ($n = 12$). The internal standard PS-d5 was used for continuous repeatability checks of sample measurements.

To evaluate the plastic recovery from soil, triplicates of two agricultural reference soils were spiked at 2 and 20 $\text{μg} \, \text{g}^{-1}$ of

each polymer. The used reference soils were a loamy sand (8 % clay, 16 % silt, 76 % sand) with a $C_{org}$ content of 1.7 % (LUFA 2.2, Landwirtschaftliche Untersuchungs- und Forschungsanstalt, Speyer, Germany) and a silty clay (47 % clay, 41 % silt, 12 % sand) with 2.5 % $C_{org}$ (RefeSol 06-A, Fraunhofer IME, Schmallenberg, Germany). Instrumental and method limits of detection (LODs) were calculated from standard deviations (SDs) of signal intensities of low analyte concentrations (2 $\text{μg} \, \text{mL}^{-1}$) and blank reference soils ($n = 3$), respectively, in accordance with DIN 32645 (2008) and Magnusson and Örnemark (2014). The

selectivity against other, potentially interfering non-target polymers was estimated from peak intensities of PE, PP, and PS pyrolysates in LUFA 2.2 soil spiked at each 40 $\text{μg} \, \text{g}^{-1}$ polyethylene terephthalate (PET), poly(methyl methacrylate) (PMMA), polyvinyl chloride (PVC), and tire wear debris (TWD). In addition, a matrix-matched calibration was performed in LUFA





2.2 soil extracts in accordance with Magnusson and Örnemark (2014) to assess the effect of soil matrix on the calibration
parameters. All measurements were monitored with procedural blanks.

## 2.7 Data evaluation

Data processing and statistical analyses were conducted using R (version 4.1.0) with "data.table", "magrittr", and "envalysis"
as main libraries. The results are given as mean ± SD. Measurement repeatabilities are stated as percentage relative standard
deviation (RSD). Matches from the Open Specy FTIR library are reported as Pearson's $r$.

The potential matrix effect on the calibration was evaluated using signal suppression/enhancement ratios (SSEs, Eq. 1)
which compare the slope of a calibration curve prepared in solvent ($b_{\mathrm{solv}}$) with that of the matrix-matched calibration ($b_{\mathrm{matrix}}$)
(Magnusson and Örnemark, 2014).

$$\mathrm{SSE} = \frac{b_{\mathrm{matrix}} - b_{\mathrm{solv}}}{b_{\mathrm{solv}}} \tag{1}$$

## 3 Results and discussion

### 3.1 Soil properties

According to FAO classification (IUSS Working Group WRB, 2015), the investigated soils were identified as anthrosols. The
dominant soil textures were silty clay and clayey silt (Table 1), with sites 1, 3, and 4 showing the highest clay contents ($\geq 30\,\%$)
compared to the remaining sites. Sites 1–3 and sites 6–7 had $C_{\mathrm{org}}$ contents of 1.1–1.3 % and 1.4–1.5 %, respectively. The lowest
$C_{\mathrm{org}}$ contents (0.9 %) were found on sites 4 and 5. Soil N was $\leq 0.2$ % across all sites. The soil pH was slighly acidic (6.6–7.0),
and the EC ranged from 118 to 536 $\mu\mathrm{S\,cm}^{-1}$. The highest EC values were observed on sites 6 and 7.

### 3.2 Agricultural plastic covers

The fleeces that covered sites 1 and 2 were identified as PP as indicated by multiple $C{-}H$ stretch deformations at 2950–
2838 $\mathrm{cm}^{-1}$ as well as $CH_2$ and $CH_3$ bends at 1455 and 1377 $\mathrm{cm}^{-1}$, respectively (Open Specy FTIR library match: $r \geq 0.96$,
see Fig. A1a for an exemplary FTIR spectrum). A shapeless broad peak between 1860 and 1660 $\mathrm{cm}^{-1}$ indicated the presence
of carbonyl groups (Grause et al., 2020). Complementary thermoanalysis showed crystallization temperatures at 114–116 °C
and melting temperatures at 158–160 °C. Between 381 and 400 °C, the polymers started to decompose into methylalkenes
characteristic for PP (Tsuge et al., 2011, Fig. A2a for an exemplary pyrogram).

By contrast, the fleece from sites 4 and 5 was of PE ($r = 0.96$). The respective FTIR spectrum showed indicative $CH_2$ stretch-
ing between 2919–2915 $\mathrm{cm}^{-1}$ (asymmetric) and 2851–2845 $\mathrm{cm}^{-1}$ (symmetric, Fig. A1c). The crystallization and melting
temperatures were 96 and 108 °C, respectively. The degradation onset was 408 °C and triggered the formation of PE-specific
triplets of $n$-alkadienes, $n$-alkenes, and $n$-alkanes (Fig A2c). All other covers, namely mulches and perforated foils from sites
2–8, were made of PE ($r \geq 0.86$, Fig. A1b,d,e). The carbonyl band at 1860–1660 $\mathrm{cm}^{-1}$ was visible in all sample but was most





**Table 1.** Soil properties of experimental sites.

| Site | Cover (bottom up) | Location | Clay [%] | Silt [%] | Sand [%] | Texture[‡] | $C_{org}$ [%] | $C_{total}$ [%] | $N_{total}$ [%] | pH | EC [$\mu S\,cm^{-1}$] |
|---|---|---|---|---|---|---|---|---|---|---|---|
| 1 | Fleece (PP), perforated foil (PE) | Offenbach | 34 | 53 | 13 | Tu3 | 1.3 | 1.3 | 0.1 | 6.6 | 200 |
| 2 | Mulch (PE), fleece (PP) | Offenbach | 10 | 77 | 13 | Ut2 | 1.1 | 1.3 | 0.1 | 6.8 | 138 |
| 3 | Mulch (PE) | Offenbach | 36 | 64 | 0 | Tu3 | 1.2 | 1.4 | 0.1 | 6.8 | 147 |
| 4 | Fleece (PE), perforated foil (PE) | Schifferstadt | 32 | 67 | 1 | Tu4 | 0.9 | 1.1 | 0.1 | 6.7 | 118 |
| 5 | Fleece (PE), perforated foil (PE) | Schifferstadt | 24 | 76 | 0 | Ut4 | 0.9 | 1.1 | 0.1 | 6.9 | 236 |
| 6 | Perforated foil (PE) | Landau | 25 | 75 | 0 | Ut4 | 1.4 | 1.6 | 0.2 | 6.8 | 510 |
| 7 | Perforated foil (PE) | Landau | 21 | 79 | 0 | Ut4 | 1.5 | 1.6 | 0.2 | 6.9 | 536 |
| 8 | Perforated foil (PE) | Landau | 15 | 85 | 0 | Ut3 | 1.4 | 1.6 | 0.2 | 7.0 | 289 |

[‡]in accordance with Sponagel et al. (2005); Tu = silty clay, Ut = clayey silt.

pronounced for the PE mulch from sites 2 and 3. However, crystallization temperatures (100–113 °C) and melting temperatures (110–122 °C) of the PE mulch were slightly higher than those of the PE fleece. The degradation onsets of the mulches and perforated foils ranged from 384 to 397 °C.

The qualitative analyses of volatile polymer additives and other polymer-associated compounds thermodesorbing from the agricultural films at 300 °C revealed three omnipresent substances (NIST08 matches >75 %): These were propyl dodecanoate (CAS 3681-78-5), oleonitrile (CAS 112-91-4), and 9-octadecenamide (CAS 301-02-0, see Fig. A2 for exemplary chromatograms). In addition, the PP fleeces from sites 1 and 2 as well as the PE perforated foils from sites 4–8 contained traces of BHT (CAS 128-37-0), a common antioxidant (Hahladakis et al., 2018). Propyl dodecanoate and oleonitrile are slip agents

probably added to agricultural plastic covers for easier spreading on site. 9-Octadecenamide is a known degradation product of hindered amine light stabilizers like Chimassorb 944 (Haider and Karlsson, 2001). No pesticides were detected in the plastic covers, probably due to the limited sensitivity of the qualitative analysis and/or their low thermal stability.

Complementary FTIR–ATR and Py-GC/MS confirmed that both plastic mulches and perforated foils were exclusively made of PE. The fleeces were of PE and PP, although PP is more common (Hamouz et al., 2011). All PE covers melted within the

range of 109–125 °C and degraded >318 °C as expected for virgin LDPE (Beyler and Hirschler, 2002). Interestingly though, melting temperatures of the PP fleeces were 5 to 10 °C lower than those of virgin PP (165–170 °C) (Beyler and Hirschler, 2002; Tocháček et al., 2019). The degradation onset was not affected by this and comparable to virgin PP (>315 °C) (Beyler and Hirschler, 2002). Decreasing melting temperatures may be a first sign of polymer aging as similarly observed after 5–20 months of temperate weathering (Tocháček et al., 2019). This is consistent with the carbonyl groups identified via FTIR which

are indicative for the photo-oxidation of polyolefins (Grause et al., 2020). In our study, fleeces and perforated foils were on the field for four months only. The mulches were applied two years before. The incipient aging concurred with the release of





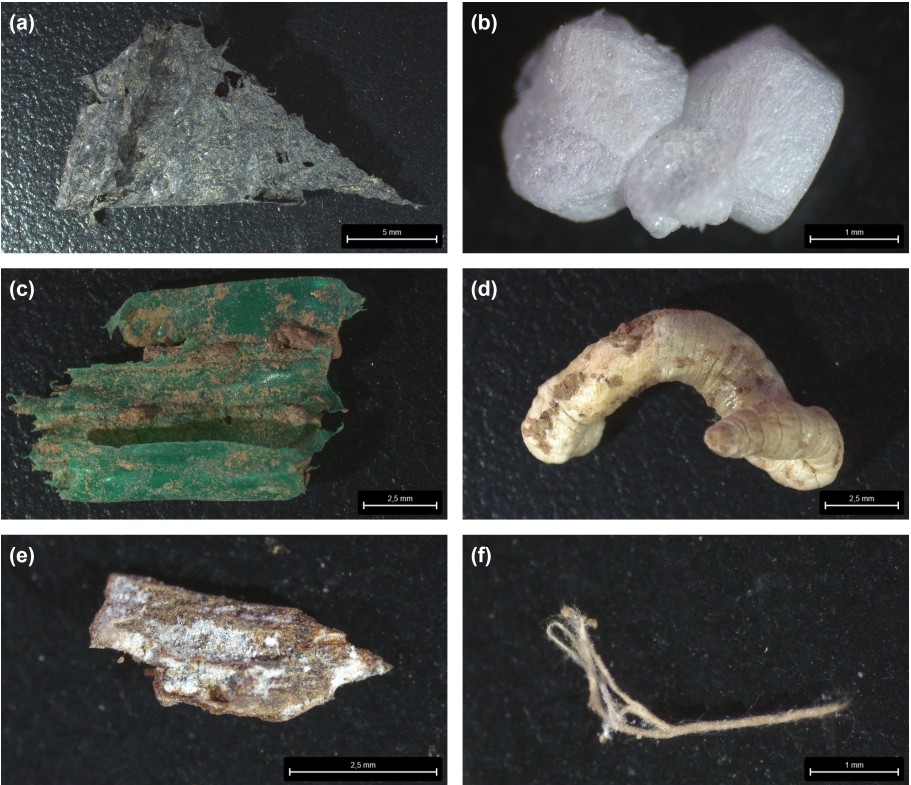

**Figure 3.** (a) PE film and (b) PS fragment from the field center of site 5, (c) PE film and (d) chitin shell from the field center of site 6, and (e) resin or natural fragment and (f) cotton fiber from the field edge of site 7; see Fig. A3 for the respective FTIR spectra.

antioxidant BHT from the PP backbone as indicated by TD-GC/MS. As BHT was also released from PE perforated foils, it remains unresolved whether the presence of BHT was material-specific or indeed triggered by polymer aging.

### 3.3 Visual plastic items on site

We visually identified 30 suspect items ($>2$ mm) during soil sieving. Subsequent FTIR–ATR analysis revealed six items as plastics. These were a black PE film ($r = 0.92$, Fig. 3a) and four PS fragments ($r \geq 0.91$, Fig. 3b) at the field center of site 5 (see Fig. A3 for the respective FTIR spectra). The PS showed characteristic peaks at 3024, 1492, and 694 cm$^{-1}$ originating from aromatic $C-H$ stretch and bend deformations. In the field center of site 6, a green PE film ($r = 0.92$, Fig. 3c) was found. All other items were of natural origin including invertebrate shells, stones or wood fragments, and cellulose fibers that were 220 identified by Open Specy as chitin, resin dispersion, and cotton, respectively ($r \geq 0.82$, Fig. 3d–f).

In this respect, it is important to note that the visual identification of suspect items largely depends on the operator's experience and may thus lead to excessive over- or underestimation of particle numbers (Thomas et al., 2020). Furthermore, counting suspect items $>2$ mm in a 100 cm$^3$ soil core is hardly representative. We thus refrained from extrapolating our findings to





**Table 2.** Instrumental validity criteria.

| Polymer | Pyrolysate | adj. $R^2$ | LOD* [ng] | RSD [%] |
|---------|-----------|-----------|-----------|---------|
| PE | 17:2(1,16) | 0.9912 | 9.0 | 11.3 |
|    | 18:2(1,17) | 0.9785 | 9.2 | 9.8 |
|    | 19:2(1,18) | 0.9965 | 5.6 | 11.3 |
|    | 20:2(1,19) | 0.9788 | 6.4 | 11.9 |
|    | 21:2(1,20) | 0.9897 | 10.0 | 12.8 |
|    | 22:2(1,21) | 0.9952 | 5.8 | 9.5 |
|    | 23:2(1,22) | 0.9709 | 7.2 | 11.2 |
| PP | 2,4Me9:1(1) | 0.9997 | 4.6 | 8.9 |
| PS | Sty | 0.9980 | 6.6 | 3.5 |
| PS | $\alpha$MeSty | 0.9866 | 19.4 | 11.5 |

*instrumental limit of detection; RSD = relative standard deviation.

particles kg$^{-1}$ and intended visual identification to serve as a qualitative complement to subsequent Py-GC/MS quantification.

Interestingly though, the plastic debris >2 mm were exclusively found on sites 5 and 6. None of the black and green PE or white PS fragments matched the applied white PE fleece and perforated foil in color or polymer type. This suggests an external source of plastic debris, for instance from adjacent streets or other fields, or residues from previous land use (Harms et al., 2021).

### 3.4 Py-GC/MS method performance

The pyrolysates chosen for PE, PP, and PS quantification were 22:2(1,21), 2,4Me9:1(1), and Sty, respectively, as they performed best in terms of signal linearity (adj. $R^2$ > 0.995), instrumental LODs (<10 ng), and measurement repeatability (RSD < 10 %, Table 2). LUFA 2.2 exerted a negligible matrix effect of 16 %, −3 %, and −2 % SSE on the selected PE, PP, and PS pyrolysates (see Table A4 for calibration curves). Methodological LODs were 0.3–0.8 µg g$^{-1}$ in RefeSol 06-A and 1.4–2.2 µg g$^{-1}$ in LUFA 2.2 (Table 3). A LUFA 2.2 soil containing each 40 µg g$^{-1}$ of potentially interfering, non-target PET, PMMA, PVC, and

TWD induced false positive intensities below the method LOD.

The extraction of 20 µg g$^{-1}$ plastic debris from LUFA 2.2 soil yielded recoveries of 86–105 % (Table 3). PE was recovered best while PS showed the lowest value. Recovering plastic debris at levels close to the method LOD (2 µg g$^{-1}$) led to an overestimation of recovered PE (133±9 %) while underestimating PP (70 %) and PS (50 %). Recoveries from RefeSol 06-A were generally lower: While we still recovered 50 and 62 % of the 20 µg g$^{-1}$ PE and PP, respectively, recoveries dropped to

30±20 % at the lower spiking level. Hardly any PS was recovered from RefeSol 06-A (<12 %) irrespective of the spiking level.





**Table 3.** Validation criteria for the extraction method.

| Polymer | Pyrolysate | LOD* [µg g$^{-1}$] | Interference† [µg g$^{-1}$] | Recovery at 2 µg g$^{-1}$ [%] | at 20 µg g$^{-1}$ [%] |
|---|---|---|---|---|---|
| *LUFA 2.2* | | | | | |
| PE | 22:2(1,21) | 1.4 | 0.9±0.3 | 133±9 | 105±3 |
| PP | 2,4Me9:1(1) | 1.6 | 0±0 | 70±10 | 93±5 |
| PS | Sty | 2.2 | 0±0 | 52±2 | 86±4 |
| *RefeSoil 06-A* | | | | | |
| PE | 22:2(1,21) | 0.8 | | 30±20 | 50±10 |
| PP | 2,4Me9:1(1) | 0.3 | | 30±20 | 62±1 |
| PS | Sty | 0.5 | | 0±0 | 12±5 |

*method limit of detection; †introduced from 40 µg g$^{-1}$ non-target polymers.

As recently reviewed by Thomas et al. (2020), several studies have already evaluated their extraction procedures for various plastic debris from solid matrices using organic solvents like dichloromethane (DCM) or tetrahydrofuran (THF). In combination with quantitative Py-GC/MS, however, matrix interferences and false positive detections from organic matrix constituents or other, non-target polymers should be closely monitored (Dierkes et al., 2019; Steinmetz et al., 2020). By combining den-

sity separation and solvent extraction with xylene/TCB, we obtained method LODs from blank LUFA 2.2 and RefeSoil 06-A and interferences from non-target polymers equivalent to 1–2 µg g$^{-1}$ PE, PP, and PS. Dispersing soil aggregates with sodium hexametaphosphate prior to density separation further enabled the quantification of plastic debris potentially occluded in or masked by soil aggregates. Based on the two reference soils tested, our method is considered sufficiently sensitive, robust, and selective for environmentally-relevant plastic levels. However, extrapolation of these validity criteria to field samples with a

different texture and $C_{org}$ composition remains difficult and requires careful interpretation.

This similarly applies to the PE and PP recoveries <30 % we obtained from the clayey RefeSoil 06-A with a $C_{org}$ content of 2.5 % at spiking levels close to the method LODs (2 µg g$^{-1}$). While this clearly defines the quantitative limit of our method, our working range is still 10–100 times lower than that of previous applications involving solvent-based Py-GC/MS (Dierkes et al., 2019; Okoffo et al., 2020). Yet, our PS recoveries were particularly low, which is in line with Wang et al. (2018) who

found comparable recoveries after density separation of nano-sized PS from a silt soil. Luo et al. (2020) and Wu et al. (2020) reasoned that SOM as well as iron and aluminum oxides effectively retain PS particles in soil. The dramatic decrease in PS recovery may be attributed to organo–mineral interactions forming between the delocalized $\pi$-electrons of the aromatic PS ring and SOM, iron and aluminum oxides, or cations bound to the negatively charged surface of clay particles (Newcomb et al., 2017). During the density separation, the aggregated PS may have been preferentially sedimented, and thereby systematically

excluded from subsequent solvent extraction. The addition of an anionic surfactant like sodium dodecyl sulfate or nonionic





polysorbates during soil aggregate dispersion and density separation could counteract this, but potentially at the expense of introducing another source of PE contamination from the surfactants' $n$-alkane domains.

Despite that, we considered the 86–105 % recovery of 20 µg g$^{-1}$ PE, PP, and PS from the sandy LUFA 2.2 soil quantitative and comparable to previous studies. Dierkes et al. (2019) and Okoffo et al. (2020), for instance, extracted various polymers
(0.05–50 mg g$^{-1}$) via accelerated solvent extraction with THF at 185 °C and DCM at 180 °C, respectively. Their recoveries from 1 g of quartz sand and biosolids were 77–128 %. By contrast, a simple batch extraction setup using TCB at 120 °C recovered 70–128 % PE, PP, and PS (250 µg g$^{-1}$) from 4 g LUFA 2.2 and RefeSol 06-A reference soils (Steinmetz et al., 2020). In comparison to those studies, the spiking levels of our present method are 2.5–1000 times lower in order to approach the anticipated plastic levels in our field samples. Moreover, we extracted plastic debris from 50 g soil to better account for the
heterogeneous distribution of plastic particles in soil. Nevertheless, the 50 % PE and 62 % PP we recovered from RefeSol 06-A suggest a rather semi-quantitative evaluation of clayey soils with a $C_{org}$ content >2.5 %. These findings once more highlight the importance of specifically testing and evaluating analytical methods for plastic analysis with various soil types (Thomas et al., 2020).

### 3.5 PE, PP, and PS debris in soil

We detected plastic debris ≤2 mm exceeding the estimated method LOD (>1 µg g$^{-1}$) in 41 out of 240 samples from all sites except site 6 (Fig. 4). This is equivalent to 17 % positive detections. Soil from sites 1–3, 7, and 8 contained the most PE (37 findings) with single detections peaking at 19 and 35 µg g$^{-1}$. Mean PE contents were the highest at the field margin of site 1 (10±10 µg g$^{-1}$) and decreased to barely detectable 1–2 µg g$^{-1}$ in the field edge and the field periphery. Furthermore, PE contents were slightly higher in the track rows (furrows) than in the plant rows (ridges) of the field centers and edges. In
comparison to that, the PE distribution on sites 2 and 3 was more uniform across transects. Nonetheless, mean PE contents were slightly higher at field centers and margins than in the periphery. Sites 4 and 5 did not contain any PE. With 4–7 µg g$^{-1}$, sites 7 and 8 showed maximum PE contents in the field periphery. Field centers, edges, and margins contained less than 2 µg g$^{-1}$ PE. On these sites, differences in the PE contents between field track and plant rows were mostly indistinct. PP was found on sites 2, 3, 4, and 7 (five findings >1 µg g$^{-1}$). The PP distribution was mostly driven by single observations of 5–10
µg g$^{-1}$ in the field centers. PS was identified three times, namely in the periphery of sites 4 and 5 and in the field edge (plant row) of site 4.

Interestingly, elevated PE contents occurred mostly on sites 1, 7, and 8 which were covered with 40 µm thick perforated foils. Sites 4–6 covered with thicker mulch films or perforated foils (50 µm) did not show any significant PE contamination. This is remarkable since the agricultural films were on site for four months only. Such short-term effects have not been reported
in scientific literature so far. Our results are yet in line with Zhang et al. (2016) who attributed elevated plastic emissions to the use of particularly thin agricultural films. In China, for instance, common film thicknesses are 6–10 µm, while EU regulations stipulate agricultural covers thicker than 20 µm (EN 13655, 2018). This may also explain why studies conducted in non-EU countries often report extraordinarily high plastic levels in soil (Liu et al., 2014), particularly after long-term use of agricultural plastics (Huang et al., 2020; Zhang and Liu, 2018).



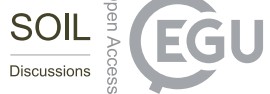

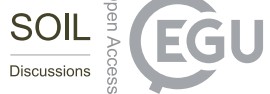

**Figure 4.** Log-scaled PE, PP, and PS contents ($\leq 2$ mm) at the field center (FC), field edge (FE), field margin (FM), and the periphery (P) of sites 1–8; dots represent single measurements, the underlying bar plot shows the transect average.

Regardless of the film thickness, the elevated PE contents at the field margin of site 1 suggested that the mechanical stress of weighing the plastic covers down with soil or digging them in favored the local formation of plastic debris. The close contact with soil and exposure to sunlight may have accelerated polymer aging and embrittlement as indicated by our complementary DSC and FTIR–ATR measurements. Due to the limited number of PE detections above method LOD, we did not find a clear indication for the further translocation of plastic debris from the ridged plant rows to lower ground furrows. Tracing

experiments by Laermanns et al. (2021), however, recently confirmed that the micro- and macrorelief of the soil surface may indeed favor the water erosion of plastic debris on a meter scale. Even at larger scales though, it remained unresolved to what extent the PE debris in the field periphery (mainly sites 7 and 8) originated from the covered field centers or whether it came from an external source via wind drift. Due to ubiquity of products made from PE, such an external source cannot be excluded.





Although sites 1 and 2 were both fleeced with PP, only site 2 showed elevated PP contents. At the same time, PP was found
on sites covered exclusively with PE for the last four months. Therefore, no clear association between PP detections and the
seasonal use of plastic covers was established. This is striking because the fibrous structure of the PP fleece together with the
initial signs of aging detected via DSC and FTIR–ATR made emissions of plastic debris particularly likely. Unexpectedly, these
two study sites were thus most likely dominated by external sources like littering or previous land use rather than receiving
plastic debris from the in situ fragmentation of fleeces.

This similarly applied to PS, which is not used for agricultural plastic covers (Bertling et al., 2021) and may thus serve as an
indicator for external sources of plastic debris in soil. Another possible explanation for the PS findings on the two neighboring
sites 4 and 5 may be a legacy contamination with PS. In the past, beads made from expanded PS were used for the conditioning
and stabilization of horticultural soils (Maghchiche et al., 2010).

All in all, the plastic contents detected in our study were up to 200 times lower than the 820 µg g$^{-1}$ PE, 40 µg g$^{-1}$ PP,
and 56 µg g$^{-1}$ PS that Dierkes et al. (2019) obtained from a non-characterized roadside soil quantified via solvent-based Py-
GC/MS. By contrast, plastic levels in floodplain soil were estimated at 5 µg g$^{-1}$ based on particle counts and sizes (Scheurer
and Bigalke, 2018). However, conversions from particle counts and sizes to masses are increasingly discouraged for their high
estimate errors (Thomas et al., 2020; Primpke et al., 2020). This challenges further comparisons since studies investigating
plastic debris in agricultural soil so far exclusively used particle-based microspectroscopic techniques.

**4   Conclusions**

The combination of soil aggregate dispersion and density separation with solvent-based Py-GC/MS enabled the simple, yet
robust quantification of PE, PP, and PS debris in agricultural soil. Analyzing a sample amount of 50 g better accounted for the
heterogeneous distribution of discrete plastic particles in the soil matrix. The additional dispersion step further made plastic
debris occluded in soil aggregates amenable to quantification.

We successfully validated and applied our new method to soil randomly sampled from four predefined transects located in
and around eight agricultural field covered with plastic films. This screening approach revealed first insights into the potential
contribution of agricultural plastic covers to plastic pollution in soil: While PP fleeces and 50 µm thick PE films were not
indicated to emit plastic debris into their surrounding during their use, four months of covering with thinner perforated PE
foils (40 µm thickness) was associated with elevated PE contents in and around the covered fields. The identified plastic levels
were below 35 µg g$^{-1}$ and with that up to two orders of magnitude lower than those previously reported for soil (Dierkes et al.,
2019). This indicates that current EU regulations (EN 13655, 2018) and recycling efforts for agricultural plastics start to take
effect. The long-term use of thin perforated foils, in particular, is however likely to contribute to the accumulation and further
distribution of plastics in the environment. To prevent this, the use of thicker and more durable plastic covers may be preferred.

To scrutinize this, future research should aim for the continuous monitoring of plastic contents in soil. This may also include
samplings of deeper soil and more sensitive screenings of polymer-associated compounds including additives and agrochemi-
cals sorbed to the plastic covers. Advancing the field of mass spectrometric methods for the quantification of plastic debris in

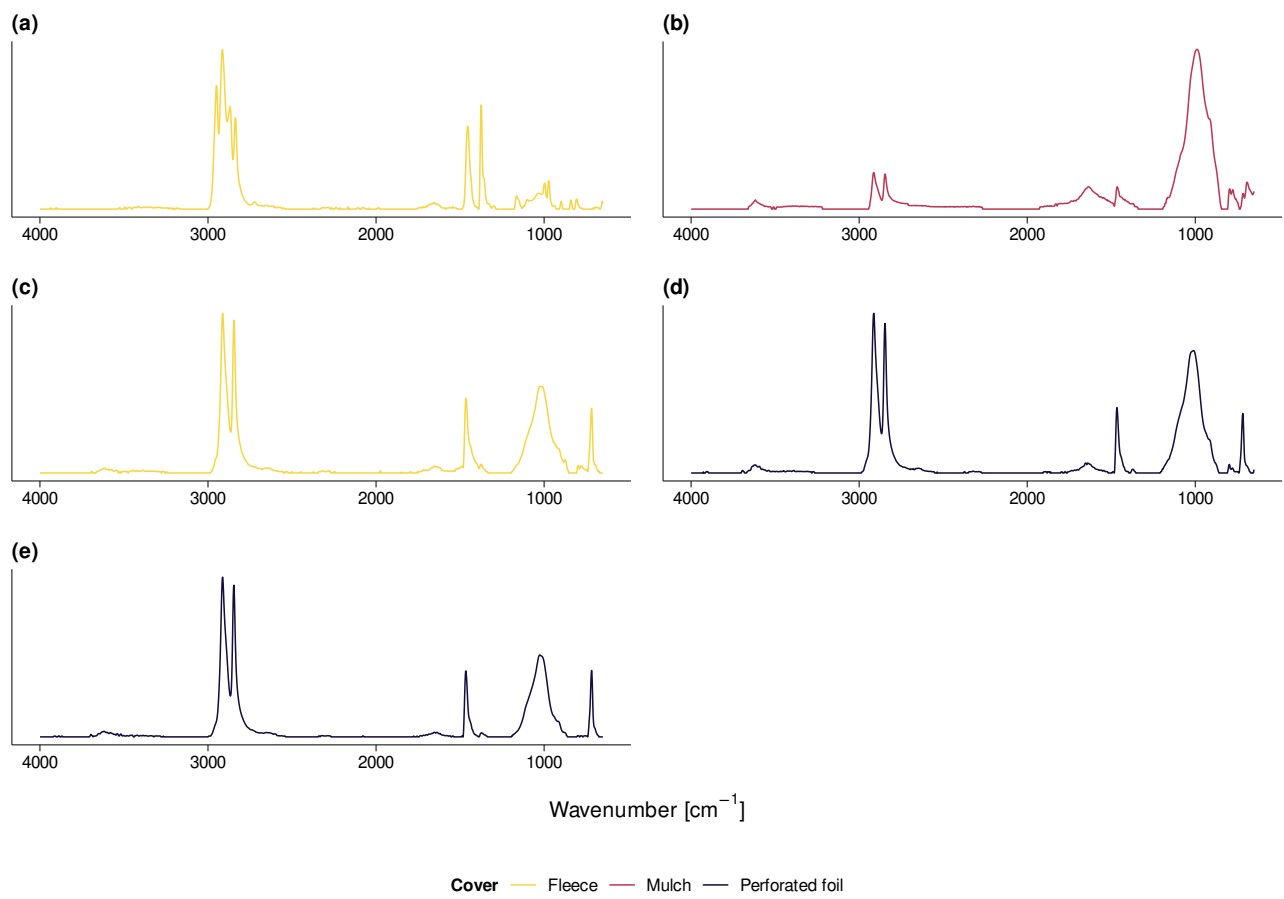

**Figure A1.** Exemplary FTIR spectra of (a) the PP fleece from site 1, (b) the PE mulch from sites 2 and 3, (c) the PE fleece and (d) PE perforated foil from sites 4 and 5, and (e) the PE perforated foil from site 8.

heterogeneous matrices will help to bridge the gap between modeling and monitoring, further the science-based regulation of agricultural plastic products, and contribute to their sustainable use.

*Code and data availability.* All data and code to reproduce data processing and statistical analysis are publicly available from https://doi.
org/10.6084/m9.figshare.14742849.





**Figure A2.** Exemplary chromatograms of polymer pyrolyses (700 °C, left) and thermodesorption (300 °C, right) of polymer additives; (a) PP fleece from site 1, (b) PE mulch from sites 2 and 3, (c) PE fleece and (d) PE perforated foil from sites 4 and 5, and (e) PE perforated foil from site 8.



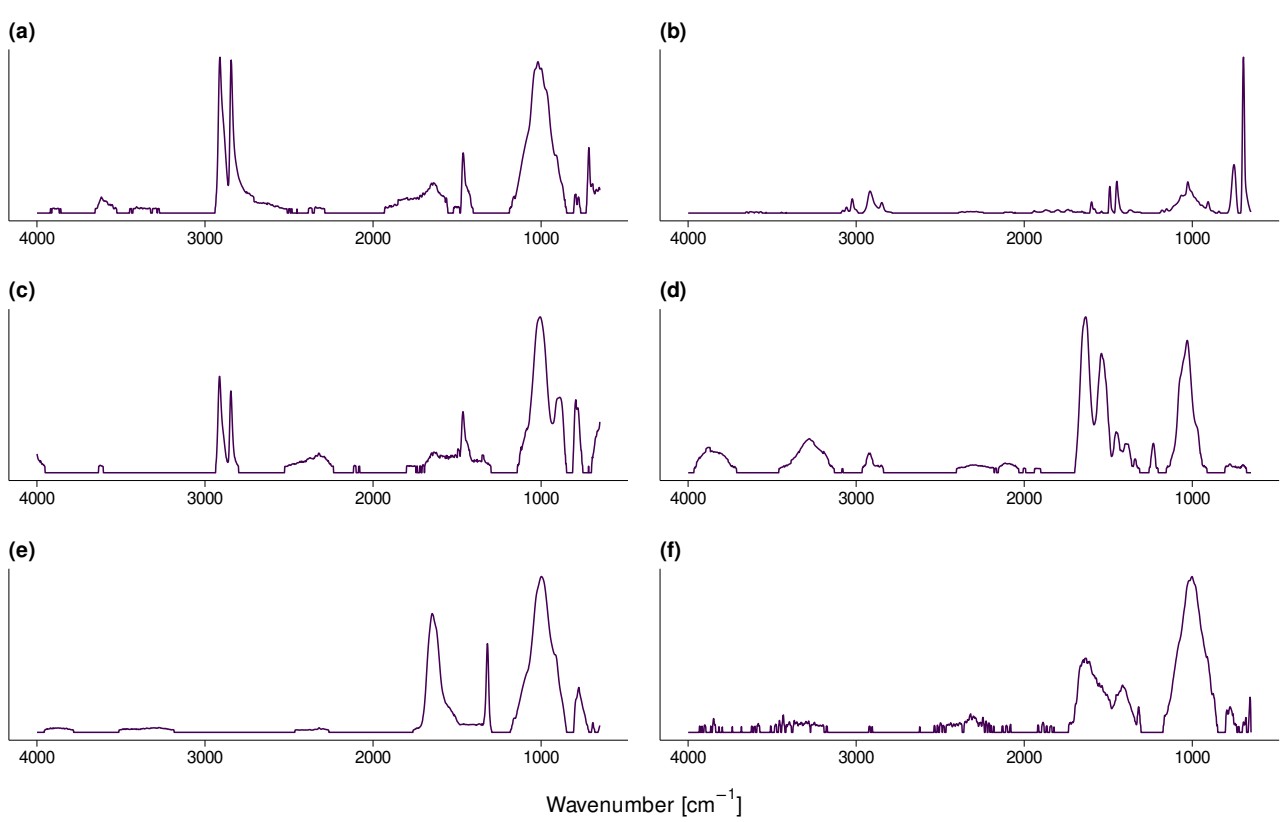

**Figure A3.** FTIR spectra of the debris shown in Fig. 3; (a) PE film and (b) PS fragment from the field center of site 5, (c) PE film and (d) chitin shell from the field center of site 6, and (e) resin or natural fragment and (f) cotton fiber from the field edge of site 7.

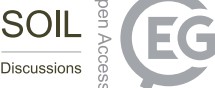

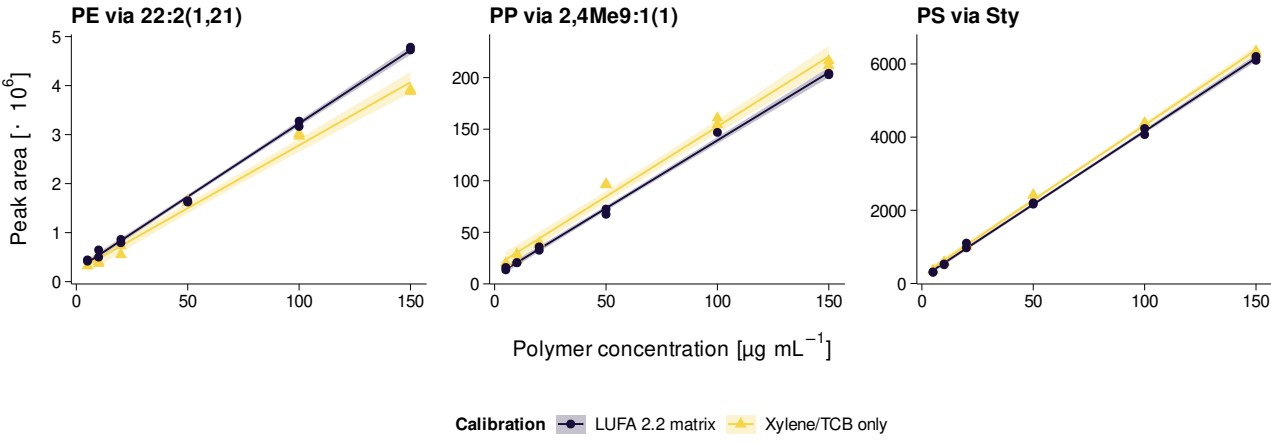

**Figure A4.** Py-GC/MS calibration in solvent and LUFA 2.2 matrix compared.

*Author contributions.* ZS conceived the idea, designed the study, performed the analyses, and took the lead in writing the manuscript. PL and SE assisted with sample extractions and Py-GC/MS and FTIR measurements. JD conducted thermogravimetric and DSC analyses. KM and GS supervised the project. The manuscript was finalized through contributions of all authors.

*Competing interests.* The authors declare no competing conflicts of interest.

*Acknowledgements.* This study was financially supported by the pilot program "Profil³" of the "Bildung Mensch Umwelt" research fund at University of Koblenz–Landau (project "PLAST") and by the Ministry for Education, Science, Further Education and Culture (MBWWK) of Rhineland-Palatinate in the frame of the Interdisciplinary Research Group for Environmental Studies (IFG Umwelt) of the University of Koblenz–Landau. Maximilian Meyer and Aaron Kintzi are acknowledged for their help with soil sampling and preliminary experiments. We further thank Wiebke Mareile Heinze for fruitful discussions and feedback on the manuscript.





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
