# Peer review of "Are agricultural plastic covers a source of plastic debris in soil? A first screening study"

_SOIL, 2021_

## Author Response (AR1)

Dear Prof. Dr. Zaccone, dear reviewers,

Thank you very much for the opportunity to revise. The very constructive reviewer and editorial comments helped us to further improve our manuscript. This includes the addition of a quality control section, the (re)calculation of method LODs and LOQs, and a rewrite of the results and discussion as suggested by reviewer #1 (RC1). We further streamlined the manuscript in accordance with the comments made by reviewer #2 (RC2).

In our revision, we addressed all reviewer comments and corrected a few minor mistakes. Please find below the detailed response to the reviewers' comments. We hope that the revised manuscript is now acceptable and convinces you and the reviewers.

Kind regards,

Zacharias Steinmetz

on behalf of all co-authors

**Detailed point-by-point responses to the revision of Steinmetz et al. (soil-2021-70)**

> Remark: The **bold blue** line numbers refer to the numbering in the clean version of the revised manuscript. Changes to the submitted version are highlighted in yellow.

**Reviewer #1 (RC1)**

*Method*

1. The authors have not included any information on laboratory quality control measures which are a must for microplastics studies. Were laboratory and field blanks analysed and how? What were concentrations in the blank samples? Were deposition blanks conducted during the FTIR analysis? Were samples extracted in a fume hood, were lab coats (what sort) worn during extraction? Were duplicates conducted to assess heterogeneity in the sample? What were your internal standard recoveries, were polymer concentrations recovery corrected? Please include a section for QA/QC in the manuscript.

In the submitted version of our manuscript, we intended to keep QA/QC brief and combined it with our method validation (see former Section 2.6: Method validation and quality control). Therein, we stated that "The soil cores were immediately transferred to uncoated paper bags and air-dried therein to reduce the risk of contamination." and "All measurements were monitored with procedural blanks.".

We expanded this to a more comprehensive QA/QC section which now reads as follows.

> **Lines 171–174**: To prevent the risk of contamination, all laboratory equipment coming into direct contact with the sample or the extract solution was made of glass, metal, paper, or PTFE. PE, PP, or PS equipment was completely avoided. The worn laboratory coats were of 100% cotton. In addition, all samples and extracts were kept in closed vessels or covered with aluminum foil. The vessels were only opened under a fume hood.

> The sample extraction was monitored with weekly procedural blanks that underwent the complete extraction procedure as the samples but without soil addition. Plastic contents in our procedural blanks were exclusively below the LOD.

Please note that we did not use field blanks because we could hardly treat them in the same way a sampled soil is. But we analyzed all our equipment, including the used paper bags, for their contamination potential and found them not interfering our analysis. In addition, it remains worth noticing that the majority of our soil samples did not contain any plastics (<LOD) which suggests negligible sample contamination.

Furthermore, we did not run FTIR deposition blanks for the particles >2mm since our FTIR–ATR analysis only aimed at the qualitative identification of single suspect particles. In this case, the FTIR signal of the sample surface was expected to significantly exceed that of dust traces on the particle.

Since we designed our study to be a first screening, we ran single measurements only. We further clarified this in the methods section.

> **Lines 137–138**: Each sample was measured once as described in Section 2.3.

We further did not assess the recovery of our internal standard, namely deuterated PS (PS-d5). This is because PS-d5 was only added after sample extraction and served as a quality control measure for internal instrumental repeatability during measurement sequences.

We put this more clearly by adding

*Lines 152–153: The internal standard PS-d5 ==added after sample extraction was== used for continuous repeatability checks of sample measurements.*

2. More information is needed on the Py-GC-MS quantification. Why were the dienes chosen for quantification of the polyethylene (PE), was this from a previous published method? Were the samples analysed in full scan or SIM mode?

Our solvent-based Py-GC/MS approach was originally published in Steinmetz et al. (2020). In order to avoid extensive repetitions, we tried to keep this short. Yet, we assessed the Py-GC/MS method performance once more in the present manuscript. We stated that "The pyrolysates chosen for PE, PP, and PS quantification were 22:2(1,21), 2,4Me9:1(1), and Sty, respectively, as they performed the best in terms of signal linearity (adj. R2 > 0.995), instrumental LODs (<10 ng), and measurement repeatability (RSD <10 %, Table 2).".

To make this clearer, we added

*Lines 246–247: ==The n-alkadiene 22:2(1,21) was preferred over the respective n-alkene or n-alkane because of its higher selectivity for PE (Steinmetz et al., 2020).==*

Furthermore, our Py-GC/MS measurements were ran in SIM mode, which we report in Line 140ff.

To further clarify this, we added "SIM mode" in parentheses.

*Lines 140–143: The MS selectively monitored ==(SIM mode)== m/zs 70 and 126 for the PP pyrolysate 2,4-dimethyl-1-heptene (2,4Me9:1(1), RI 841), m/zs 104 and 118 for the PS pyrolysates styrene (Sty, RI 895) and α-methylstyrene (αMeSty, RI 981), respectively, and m/zs 82 and 95 for PE n-alkadienes like 1,21-docosadiene (22:2(1,21), RI 2187).*

3. Styrene is not an ideal pyrolysis product for monitoring polystyrene (PS) as it is not selective. It can originate from organic material (although this may have been removed in your TD analysis) as well as being a pyrolysis product of many other polymers. Typically, the dimer or trimer or polystyrene is monitored. This will increase the MDLs but improve your selectivity. Also, how can you be certain the PS isn't a sampling/analysis artefact without any blank information? Combined with the poor matrix spike recoveries of PS in the

reference soil, your method is not optimised or validated for analysis of PS and you cannot confidently report these results.

Our blank chromatograms, these were the weekly procedural blanks (see item 1 above), did not contain styrene at intensities exceeding the LOD; nor did the analyses of our reference soils. In this regard, the selectivity of our method for PS does not originate from choosing styrene as a marker but (1) from the density separation excluding plastics with a density >1.2 g cm$^{-3}$ and (2) the subsequent selective dissolution of our target polymers with trichlorobenzene/xylene. We think that this is also the reason why tire wear added to our reference soil at a level twice as high as our highest standard did not induce styrene signals that exceeded the LOD.

Our approach is further in line with Fabbri et al. (2020, doi: 10.1016/j.jaap.2020.104836) who similarly used styrene as a marker compound after polymer dissolution with toluene. The authors argued that dimers may also originate from secondary reactions of monomers with one another, which would challenge their selectivity in general. Such secondary reactions are, however, disfavored when PS is spread on a thin layer or on quartz filters after the solvent has dried. After polymer dissolution, the peak intensities of the PS oligomers are thus considerably lower than those obtained after the pyrolysis of solids. Although this is a very interesting observation, we are reluctant to add it to our discussion as it deviates from the common theme of the manuscript and was already addressed by Fabbri et al. (2020). But if you and the editor prefer to have this added, we will be happy to do so.

We rather suspect the poor PS recoveries from clay soil to originate from aromatic PS domains interacting with soil particles during the density separation. This is discussed in Section 3.4: "The dramatic decrease in PS recovery may be attributed to interactions forming between the delocalized π-electrons of the aromatic PS ring and SOM, iron and aluminum oxides, or cations bound to the negatively charged surface of clay particles (Newcomb et al., 2017).".

In line with your suggestion, we now interpret the PS results more carefully. This includes the following additions/modifications to our discussion.

*Lines 272–273:* *Irrespective of the spiking level though, our PS recoveries from the clayey RefeSol 06-A were particularly low (<12 %).*

*Lines 283–285:* ==The 50 % PE and 62 % PP== *we recovered from RefeSol 06-A suggest a rather semi-quantitative evaluation of soils* ==with a clay content >47 % and a Corg content >2.5 %.== ==PS is evaluated qualitatively for its low recoveries.==

*Lines 298–299:* ==Due to the poor PS recoveries, these findings are most likely underestimated.==

We modified the conclusions accordingly:

*Lines 341–345: The combination of soil aggregate dispersion and density separation with solvent-based Py-GC/MS enabled the simple,* ==yet selective quantification of PE and PP== *debris in agricultural soil. Analyzing a sample amount of 50 g better accounted for the heterogeneous distribution of discrete plastic particles in the soil matrix. The additional dispersion step further made plastic debris occluded in soil aggregates amenable to quantification.* ==By contrast, poor PS recoveries potentially induced by that additional separation step challenged a reliable PS quantification.==

4. Details on PET, PMMA and PVC standards need to be included. What were your tyre wear debris? Were these obtained from a chemical standards company, were they prepared in house and from what type of tyres? Did you really not see a styrene peak from pyrolysis of PVC or from the styrene-butadiene rubber in tyre tread? This suggests your analysis or extraction method is not optimised.

We used the same polymers in Steinmetz et al. (2020) and thus refrained from explaining them in detail.

We added the following explanation to the revised version of our manuscript.

*Lines 162–166:* ==The PET came from a cryomilled bottle recyclate (PETKA CZ, Brno, Czech Republic) as detailed in David et al. (2018). The PMMA was ground from a commercial plexiglass provided by Bundesanstalt für Materialforschung und -prüfung (Berlin, Germany). The PVC was purchased from Aldrich Chemistry (Taufkirchen, Germany), and TWD was from a test rig at Bundesanstalt für Straßenwesen (Bergisch Gladbach, Germany).==

As detailed in our response to item 3, our solvent-based Py-GC/MS approach was selective not only because of choosing specific pyrolysis markers but also due to the density separation (1.2 g cm$^{-3}$) and selective dissolution with trichlorobenzene and xylene

that specifically targeted PE, PP, and PS. PVC and tire wear did not interfere with our analysis because they have a higher density and do not dissolve in the applied extraction mixture.

5. Your samples are filtered at 4 um. Can you comment on possibility of micro/nanoplastics in the smaller size range that may have been missed.

Thank you for this important remark. This is a common challenge of current sample preparation methods for the analysis of microplastics in complex matrices. Particles smaller than 4 µm cannot be assessed quantitatively as they will partly flush through the filter but may at a certain stage be retained when the filter becomes increasingly clogged with clay particles. Furthermore, aggregated or coated nanoplastics may be retained more efficiently than virgin ones.

We now highlight this drawback in our discussion.

**Lines 265–266:** *The required filtration step, however, systematically excluded particles <4 µm that were not retained by the used cellulose filter.*

*Validation*

1. You cannot state your method is validated for plastics in soils when one of your two soil reference materials returned unacceptably low recoveries. Further, your LODs (MDLs) are the concentration where you have acceptable method extraction and analysis recoveries. Considering you have <30% recovery for a 2 ug/g spike in the second reference soil, the method LODs certainly are not 0.3-0.8 ug/g. The extraction method needs further assessment to determine which types of soils are applicable and what the actual MDLs are. I also suggest removing PS from the analysis due to the above mentioned issues.

We agree that this needs further clarification. In the submitted version of our manuscript, we highlighted that the ".. extrapolation of these validity criteria to field samples with a different texture and Corg composition remains difficult and requires careful interpretation.". This is a general shortcoming of soil analyses since reference soils will always differ from real soil samples.

For the calculation of LODs, we adhered to the German standard DIN 32645 (2008) and the EURACHEM guideline (Magnusson and Örnemark, 2014) which define the LOD as the

minimum amount qualitatively detectable in a blank soil. In this sense, a low recovery close to the LOD (2 mg kg⁻¹) is not surprising nor contradictory.

We thus added LOQs to Table 3 and critically discussed this data throughout the manuscript. Note that, according to DIN 32645, the calculation of LOQs is an iterative process that uses the LOD as an initial value but optimizes mostly toward the calibration standards. This is why the LOQs are quite similar in both soils.

**Table 3:**

| Polymer | Pyrolysate | LOD* $[\mathrm{mg\,kg^{-1}}]$ | LOQ* $[\mathrm{mg\,kg^{-1}}]$ | Interference[†] $[\mathrm{mg\,kg^{-1}}]$ | Recovery at 2 $\mathrm{mg\,kg^{-1}}$ [%] | at 20 $\mathrm{mg\,kg^{-1}}$ [%] |
|---|---|---|---|---|---|---|
| *LUFA 2.2* | | | | | | |
| PE | 22:2(1,21) | 1.9 | 9.5 | 0.9±0.3 | 133±9 | 105±3 |
| PP | 2,4Me9:1(1) | 2.9 | 2.9 | 0±0 | 70±10 | 93±5 |
| PS | Sty | 3.3 | 6.2 | 0±0 | 52±2 | 86±4 |
| *RefeSoil 06-A* | | | | | | |
| PE | 22:2(1,21) | 1.2 | 9.5 | | 30±20 | 50±10 |
| PP | 2,4Me9:1(1) | 0.8 | 2.5 | | 30±20 | 62±1 |
| PS | Sty | 0.7 | 6.2 | | 0±0 | 12±5 |

*method limits of detection and quantification; [†]introduced from 40 $\mathrm{mg\,kg^{-1}}$ non-target polymers.

**Line 249:** The respective method LOQs ranged from 2.5 to 9.5 mg kg–1 (Table 3).

**Lines 253–255:** Recovering plastic debris at levels close to the method LOD (2 mg kg–1 ) and below the respective method LOQs led to an overestimation of recovered PE (133±9 %) while underestimating PP (70 %) and PS (50 %).

We further recalculated method LODs directly from the peak intensities of the blank soil. In the first version of our manuscript, we estimated them from averaged soil contents. This now leads to about 1.5 times higher method LODs than before. The slightly elevated LODs reduce the total number of positive detections to 15 which, however, does not affect the outcome of our study.

We now discuss LODs and LOQs in more detail.

**Lines 267–268:** Inconsistent recoveries at a spiking level below the method LOQs of 2.5–9.5 mg kg–1 challenged the sensitivity and robustness of our solvent-based approach.

**Lines 282–285:** Based on the two reference soils tested and on previous work (Steinmetz et al., 2020), we considered our method sufficiently sensitive and quantitative for environmentally-relevant PE and PP levels exceeding the respective method LOQs. The 50 % PE and 62 % PP we recovered from RefeSol 06-A suggest a rather semi-

*quantitative evaluation of soils with ==a clay content >47 % and a== Corg content >2.5 %. ==PS== ==is evaluated qualitatively for its low recoveries.==*

We hope that these changes will facilitate the interpretation of our data.

Since the primary aim of our study was to conduct a first screening of agricultural soil, we also limited our reference soils to those of agricultural origin.

2. Line 248 needs to be rewritten, as highlighted above, your method is not sensitive, robust or selective. Similarly, Lines 252-254 needs to be rewritten as I would argue your MDLs are definitely not 1-100 times lower than previously published studies.

We agree that we used "robust" in a wrong context since the performance of our method depends on the analyzed soil.

We thus modified the mentioned lines accordingly and moved the text passage to the end of the paragraph.

**Lines 282–288:** *==Based on the two reference soils tested and on previous work (Steinmetz== ==et al., 2020),== we considered ==our method sufficiently sensitive and quantitative for== ==environmentally-relevant PE and PP levels exceeding the respective method LOQs.== ==The== ==50 % PE and 62 % PP== we recovered from RefeSol 06-A suggest a rather semi-quantitative evaluation of soils with ==a clay content >47 % and a== Corg content >2.5 %. ==PS== ==is evaluated qualitatively for its low recoveries.== These findings once more highlight the importance of specifically testing and evaluating analytical methods for plastic analysis with various soil types (Thomas et al., 2020). ==The extrapolation of specific validity criteria== ==to field samples with a different texture and Corg composition thus remains difficult and== ==requires careful interpretation.==*

*Results*

1. If you didn't find any evidence of the plastic covers in the >2mm size fraction, how can you know the PE and PP detected in the <2mm size fraction are from the covers? There is not enough data to make the conclusion that the edge of the sheets are the source of the PE and PS detected on the edges of the field. Are there other common farming sources of the three plastics analysed e.g. tractors/farming equipment? Fertiliser bags? Can these sources be discounted from the study areas?

This study aimed at screening commercially managed agricultural fields for plastic debris. With this, we depended on the reports made by the respective farmers. To our knowledge, fertilizer bags were not used. However, we cannot exclude other potential sources. To address this uncertainty, we already discussed that ".. this suggests an external source of plastic debris, for instance from adjacent streets or other fields, or residues from previous land use (Harms et al., 2021)." We further reason that "Even at larger scales though, it remained unresolved to what extent the PE debris in the field periphery (mainly sites 7 and 8) originated from the covered field centers or whether it came from an external source via wind drift. Due to ubiquity of products made from PE, such an external source cannot be excluded."

Yet, we now communicate the uncertainty of our results in a clearer way.

> **Lines 327–328:** *In the past, beads made from expanded PS were used for the conditioning and stabilization of horticultural soils (Maghchiche et al., 2010). However, it remained unresolved whether this was the case for the agricultural field investigated in this study.*

2. I would suggest the low detection and variable PS results are due to the extraction method not performing for clay type soils (which are most of the sites). Did the soil type differ between the field and the periphery where the PS was detected? Again, it would be good to have field blank information here and confirmation from another PS pyrolysis product.

The screened soils have a clay content of 15–36 % which ranges between that of the two reference soils (8 and 47 % clay). For this reason, we expected that the method will perform within this range. Please see also our response further above addressing PS pyrolysis products and blanks.

We added the following to

> **Lines 283–285:** *The 50 % PE and 62 % PP we recovered from RefeSol 06-A suggest a rather semi-quantitative evaluation of soils with a clay content >47 % and a Corg content >2.5 %. PS is evaluated qualitatively for its low recoveries.*

> **Lines 329–330:** *Given that our investigated soils had a clay content of 15–36%, the obtained PE, PP, and PS contents were potentially underestimated by a factor of 1.5–2.*

3. There is not enough data to state that PE detected at sites 1,7,8 are from the perforated foils and there is not enough data to make the conclusion that application of a foil for 4 months results in detectible PE microplastics in the soil (Line 289).

We agree that we should more clearly address the uncertainty of our results and added the following to

**Lines 302–303:** *On the one hand, this is remarkable because the agricultural films were on site for four months only. On the other hand, the elevated plastic contents may have originated from another, potentially diffuse input source prior to plastic coverage.*

*Conclusions*

1. The method is not robust, as it does not have high recoveries for different soil types. Also, the method is not successfully validated as described above.

We used "robust" in the wrong context here and removed it from this sentence.

**Lines 341–345:** *The combination of soil aggregate dispersion and density separation with solvent-based Py-GC/MS enabled the simple, yet selective quantification of PE and PP debris in agricultural soil. Analyzing a sample amount of 50 g better accounted for the heterogeneous distribution of discrete plastic particles in the soil matrix. The additional dispersion step further made plastic debris occluded in soil aggregates amenable to quantification. By contrast, poor PS recoveries potentially induced by that additional separation step challenged a reliable PS quantification.*

2. As discussed above I disagree with the statement that 4 months of covering with thinner perforated foils is associated with elevated PE content as there is no evidence that the PE originated from the foil and not other sources.

We would like to emphasized that this linkage does not indicate a causal relationship. To clarify this, we added the following to

**Lines 350–352:** *Due to the ubiquitous use of plastic covers and potentially interfering external plastic sources, a causal relationship between the use of plastic covers and*

*elevated plastic levels in soil needs yet to be shown, for instance, by conducting more controlled and systematic experiments.*

*Specific comments*

Line 90: thermodesorption should be thermal desorption

Thank you for this remark. We corrected this throughout the manuscript. See, for instance

**Lines 90–91:** *The grab-sampled plastic covers were characterized by qualitative thermal desorption (TD)- and Py-GC/MS, differential scanning calorimetry (DSC), thermogravimetry (TGA), and FTIR–ATR analysis.*

Line 113: How were the soil cores homogenised?

The soil cores were sieved as a whole and homogenized manually directly after. We added this information as follows.

**Line 113:** *All soil cores were sieved to fine soil (≤2 mm) and manually homogenized as suggested by Thomas et al. (2020).*

Line 199: Please expand BHT and please include the spectral matches as a Figure for all the NIST library identified compounds from the TD analysis.

We wrote out BHT throughout the manuscript. We further added the following figure for the comparison of spectral matches to the appendix.

**Figure A3:**

[Figure]

Line 200: Do you have any reference for propyl dodecanoate and oleonitrile being added to agricultural plastic covers?

> The cited reference (Hahladakis et al., 2018) only provides general information on common polymer additives. Polymer additives of specific commercial products like agricultural covers are typically a trade secret and have to our knowledge not been published yet. If you have more detailed insights, we would be happy to have your support.

Line 206: The lower melting temperatures of PP covers (than virgin PP) may indicate addition of additives or impurities to the PP covers.

> This is interesting. We modified the sentence accordingly.
>
> > **Lines 221–222:** *Decreasing melting temperatures may* *indicate the presence of additives or other impurities but could also* *be a first sign of polymer aging as similarly observed after 5–20 months of temperate weathering (Tocháček et al., 2019)*

Line 310: Have these PS beads been used in Germany? Do you know if they were applied to these sites?

> We do not know for sure. This is why we discuss different possibilities here.

Line 315: What size range did the previous studies use and how do they compare to your study (4um-2mm).

Solvent-based Py-GC/MS methods are still a new and emerging field. To our knowledge, other solvent-based approaches have not yet been combined with density separation. While density separation allows for higher sample amounts to be analyzed (50 g), it requires subsequent filtration which may systematically exclude smaller particles. Dierkes et al. (2019) directly extracted 1 g of soil with ASE. The soil was not sieved and no lower size cutoff was reported for the used ASE filters/membranes. Primpke et al. (2020) used filters with a pore size of 1 μm for the quantification of microplastics in sediment and water. The authors, however, directly analyzed the crushed filters without dissolving the polymers. For these reasons, detailed comparisons are difficult to draw at the current stage.

Figure A1: Please overlay the reference spectra with the samples for comparison

We modified the **Figure A1** as suggested:

[Figure]

We further applied the same modifications to Figure A4:

[Figure]

Spectrum match ▢ Chitin/cotton ▢ PE ▢ PS ▢ (Natural) resin

**Reviewer #2 (RC2)**

*General*

Very welcome is the point, that the authors says, that this is a „first screening" and not a final result (including a worldwide calculation) for the rest of the world. Therefore I would avoid the deeper comparison to other studies (especially to Dierkes 2019), especially when other techniques were used. Ever through these authors did not interpret their work as a „snapshot", the goal of this article should be this „first screening". Furthermore the study should focus on the results and not on the comparison of the methods (Who is the best one?). Therefore, please shortens the text between L. 263-274, nobody needs this „Hunt for the lowest LOD" any more.

Thank you for this remark. Following your recommendation, we substantially shortened the paragraph.

*Lines 269–272: While this clearly defines the quantitative* limits of the method, our working range is still 10–100 times lower than that of previous applications involving solvent-based Py-GC/MS. Dierkes et al. (2019) and Okoffo et al. (2020), for instance, spiked 1 g of quartz sand and biosolids at 0.05–50 g kg–1 of various polymers to evaluate their accelerated solvent extraction with THF and DCM, respectively.

*Specific comments*

Is there an meaningful reason for separation of particles larger than 2 mm and subsequent analysis using ATR-FTIR or should it be better to go down with the limit value to 0,5 mm for example?

> Our solvent-based Py-GC/MS method was intended as a first and simple screening tool for soil-associated plastic debris that complies with the definition of fine soil (<2 mm). FTIR–ATR was used as a complement for all remaining larger particles.

The advantage of the present method (density separation, polymer extraction and detection) in comparison to the method of Dierkes (polymer extraction and detection) is the investigation of a higher field sample volume. Therefore it is expected, to get a more homogeneous, representative result. Did the authors proved this by various loading of sample volume with spiked polymers? Otherwise, please comment this more clearly and highlight this as a advantage from the beginning (not in the conclusion!)

> We did not assess various sample amount for sample homogeneity but simply aimed for a maximum sample size from the beginning on.
>
> To acknowledge this, we moved this to the introduction instead of a results.
>
> > **Lines 48–50:** *To better account for the heterogeneous distribution of plastic debris in soil, we further refined and validated a new sample preparation procedure involving soil aggregate dispersion and density separation* *that allowed for the analysis of up to 50 g soil.*

L.84: Please check, if paper bags contain PS signals. PS copolymers are often used for paper stabilisation and might be a source for the unclear PS signals. The reason in L 314 is very speculative and should be deleted, so far this is not documented for the investigated soil.

> Thank you for this important remark. We measured the paper bags with our solvent-based method. An extract with 250 µg/mL paper in trichlorobenzene/xylene, which exceeded the maximum concentration of our calibration curve, did not induce any PS or PE and PP signals above LOD. We agree that respective discussion is rather speculative. However, such information may be a good starting point for other researchers to conduct a follow-up study on this issue.

We thus modified the sentence to better communicate the underlying uncertainty of our statement:

*Lines 326–328: In the past, beads made from expanded PS were used for the conditioning and stabilization of horticultural soils (Maghchiche et al., 2010). However, it remained unresolved whether this was the case for the agricultural field investigated in this study.*

L.104-110: No data from MS of TGA/MS are presented. Therefore please delete this as an information (just TGA).

We changed TGA/MS for TGA throughout the manuscript and modified Lines 104–109 as follows.

*Lines 104–109: DSC and TGA measurements were conducted in accordance with David et al. (2018). In brief, DSC was applied between –50 and 250 °C (10 K min–1 ramp, 50 mL min–1 N2 flow, Q1000, TA Instruments, New Castle, US) to determine the melting and crystallization temperatures of the agricultural plastic films. For the determination of polymer degradation onsets, plastic samples were subjected to TGA (STA 449 F3 Jupiter, Netzsch, Selb, Germany). The heating ramp was 5 K min–1 from 40 to 1000 °C under a 20 mL min–1 Ar flow. The degradation onset was determined by the temperature at which the polymer starts to thermally decompose (<1 % mass loss).*

L.160-162: No results from measurements using PET, PMMA, PVC, TWD are given. Please delete this information.

PET, PMMA, PVC, and TWD were used to assess whether particularly high contents of those four polymers interfere with the selective quantification of our target polymers PE, PP, and PS. The results are reported in Line 244ff and Table 3 (Interference).

We rephrased this sentence to make the main message clearer:

*Lines 249–251: A LUFA 2.2 soil containing each 40 mg kg–1 of potentially interfering, non-target PET, PMMA, PVC, and TWD did not induce significant false positive detections of PE, PP, or PS.*

L.184: Thermoanalysis include DSC, TGA and also others methods (DMA, Rheology etc.). In this present meaning it is related to DSC. Make this more precise, to avoid misunderstanding.

Corrected.

*Lines 195–198: ==Complementary DSC== showed crystallization temperatures at 114–116 ˚C and melting temperatures at 158–160 ˚C. Between 381 and 400 ˚C, the polymers started to decompose into methylalkenes characteristic for PP (Tsuge et al., 2011, Fig. A2a for an exemplary pyrogram).*

L.185, 193: Please define what the meaning of degradation onset's means. Include the determination of this value.

We added the following explanation.

*Lines 108–109: ==The degradation onset was determined by the temperature at which the polymer starts to thermally decompose (<1 % mass loss)==.*

L.199, 212: BHT is not a common additive for polymers. It is as a antioxidant to small and tends to migrate from the polymer bulk. The observed signal is probably related to the thermal decomposition product (or because of degradation process) of Irgafos 168 or Irganox 1010, etc.

Thank you for this important remark. We implemented your suggestion as follows.

*Lines 210–212: In addition, the PP fleeces from sites 1 and 2 as well as the PE perforated foils from sites 4–8 contained traces of ==a di-tert-butylphenol (for instance CAS 96-79-4) which is an indicator for antioxidants== (Hahladakis et al., 2018).*

L.199: please replace slip agent by lubricant, is more common.

Corrected.

*Lines 212–213: Propyl dodecanoate and oleonitrile are ==lubricants== probably added to agricultural plastic covers for easier spreading out on site.*

L.201-202: The Non-identification of pesticides are very surprisingly: Are they used during the period of agriculture? Or are they expected as a additive in the plastic materials?

> To our knowledge, the investigated covers were not marketed with added pesticides. However, it is common practice to continue pesticide applications while the agricultural covers are in place. For that reason, we also wondered why we did not find traces of those pesticides on the covers. We discuss that this was ".. probably due to the limited sensitivity of the qualitative analysis and/or their low thermal stability.". Yet, the screening of pesticides was not the primary goal of our study but we found this an interesting side note worth sharing with the scientific community.

L. 215pp: please comment the very intensive signals about 1000 cm-1 in the ATR-FTIR spectra, which are not related to PE or PP.

> We added the following sentences for clarification.

> > **Lines 194–195:** The indistinct band between 1200 and 900 cm−1 may be attributed to C–O stretching in alcohols, acids, or ethers originating from a contamination with SOM or plastic aging (Fu et al., 2021).

> Used literature:

> Fu, Q., Tan, X., Ye, S., Ma, L., Gu, Y., Zhang, P., Chen, Q., Yang, Y., and Tang, Y.: Mechanism Analysis of Heavy Metal Lead Captured by Natural-Aged Microplastics, Chemosphere, 270, 128 624, https://doi.org/10.1016/j.chemosphere.2020.128624, 2021.

L.230pp: The low recovery rate of PS is surprisingly, because PS is the best soluble polymer compared to PE and PP. The explanation in l. 257 is very speculative and needs a approval. I expect, that part of the PS degrades to smaller PS oligomers or monomers, which does not fit to the calibration signals of reference measurements. PS is very sensitive to depolymerisation (Ceiling Temperature!), therefore I expect a significant lost of signals due to degradation during density separation and extraction processes.

> In our previous study (Steinmetz et al., 2020), we applied a similar solvent-based Py-GC/MS approach on the same reference soils but without prior density separation. We obtained PS recoveries ranging from 77 to 119 %. Given that we also dissolved our PS standards prior to Py-GC/MS analysis, we assumed the additional density separation to

be the driving factor for the low PS recovery. However, we used saturated NaCl solution for density separation which we expected to have a negligible influence on PS depolymerization. From our perspective, this leaves polymer–mineral interactions as the most likely explanation which is also in line with the discussed literature.

L.275pp: Are all measurements using PY-GC/MS realised only once, or are there repetitive measurements at single samples?

Since we designed our study to be a first screening, we ran single measurements only. We further clarified this in the methods section.

**Lines 137–138:** *Each sample was measured once as described in Section 2.3.*

---

## Author Response (AR2)

Dear Prof. Dr. Whitaker, dear Prof. Dr. Zaccone,

Thank you very much for the opportunity to publish with SOIL. We would prefer to leave the additional figures in the main manuscript.

Kind regards,

Zacharias Steinmetz

on behalf of all co-authors